# Imitating Cost Constrained Behaviors in Reinforcement Learning

**Primary Keywords:** *(2) Learning;*

## Abstract

Complex planning and scheduling problems have long been solved using various optimization or heuristic approaches. In recent year, imitation learning that aims to learn from expert demonstrations has been proposed as a viable alternative in solving these problems. Generally speaking, imitation learning is designed to learn either the reward (or preference) model or directly the behavioral policy by observing the behavior of an expert. Existing work in imitation learning and inverse reinforcement learning has focused on imitation primarily in unconstrained settings (e.g., no limit on fuel consumed by the vehicle). However, in many real-world domains, the behavior of an expert is governed not only by reward (or preference) but also by constraints. For instance, decisions on self-driving delivery vehicles are dependent not only on the route preferences/rewards (depending on past demand data) but also on the fuel in the vehicle and the time available. In such problems, imitation learning is challenging as decisions are not only dictated by the reward model but are also dependent on a cost constraint model. In this paper, we provide multiple methods that match expert distributions in the presence of trajectory cost constraints through: (a) Lagrangian-based method; (b) Meta-gradients to find a good trade-off between expected return and minimizing constraint violation; and (c) Cost-violation-based alternating gradient. We empirically show that leading imitation learning approaches imitate cost-constrained behaviors poorly and show that our meta-gradient-based approach achieves the best performance.

## Introduction

Complex planning and scheduling problems have long been studied in the literature using a wide variety of optimization algorithms and heuristics. One generic way to think about the process of solving these complex problems is to first define a set of states, where a state represents the minimal amount of information required to describe a planning or scheduling instance; then for each state, compute the recommended action by optimizing certain well-defined performance measure. A comprehensive list of (state, action recommendation) tuples is then called the policy for the solved problem, and can be used in practice easily by looking up the encountered states.

An alternative paradigm to the above classical approach is to directly "learn" from an expert. More specifically, with a few expert-demonstrated traces of actions, try to generalize and derive actionable policies. Such an approach is generally called "imitation learning", which aims to replicate expert behaviors by directly observing human demonstrations, eliminating the need for designing explicit reward signals as in reinforcement learning (RL) (Abbeel and Ng 2004). This approach has been successfully applied in a variety of domains such as robotics (Fang et al. 2019), autonomous vehicles (Kuefler et al. 2017), game AI (Hussein et al. 2017), and more recently scheduling (Ingimundardottir and Runarsson 2018). Typically, this is achieved through techniques such as behavioral cloning (Bain and Sammut 1995), inverse reinforcement learning (Ng, Russell et al. 2000), and generative inverse reinforcement learning (GAIL) (Ho and Ermon 2016).

The previous research in the fields of imitation learning and inverse reinforcement learning has primarily concentrated on mimicking human behaviors in unconstrained environments, such as mimicking driving a vehicle without any limitations on fuel consumption by the vehicle. However, in many practical planning and scheduling scenarios, experts consider not only rewards or preferences but also limitations or constraints. For example, the decisions made by a self-driving delivery vehicle are not only based on route preferences or rewards, which are derived from past demand data but also on the amount of fuel/power available in the vehicle. As another example, when an agent is being trained to drive a car on a race track, the expert demonstrations that the agent is mimicking involve high-speed driving and precision maneuvering, which are critical for success in a race. However, it is also essential for the agent to adhere to safety constraints, such as staying within the track's boundaries and avoiding collisions with other vehicles. These safety constraints differ from the reward function, which may focus on achieving a fast lap time or winning the race. Therefore, the agent must strike a balance between the goal of imitating the expert demonstrations and the need to adhere to the safety constraints in order to successfully complete the task.

In scenarios where the decision-making process is influenced by both a reward model and a cost constraint model, the implementation of imitation learning becomes significantly more complex. This is because the decisions made are not solely based on the reward model, but also take into consideration the limitations imposed by the cost constraint

model. To that end, we provide a new imitation learning problem in cost-constrained environments.

Our work is closely related to Malik et al. (2021) and Cheng et al. (2023). In Malik et al. (2021), cost constraints have to be learned from expert trajectories in scenarios where the reward function is already specified. On the other hand, in Cheng et al. (2023), both safety constraint (maximum cost limit) and cost signals are known and provided, while the reward signal remains undisclosed.

The major differences of our proposed work to the above two past studies are that we assume cost signals (from sensors) from the environment are known and provided, but the safety constraint (maximum cost limit) is not given. In other words, we assume that we have necessary sensors to monitor cost signals (e.g., battery level and temperature of an EV), yet we do not know the constraints on these sensor values (e.g., safety ranges on battery level and temperature).

In terms of the solution approach, our work is similar to Cheng et al. (2023) in that we also utilize the combination of the Lagrangian-base method and the GAIL framework (Ho and Ermon 2016). However, since we do not assume knowledge of the safety constraint, we cannot include the maximum cost limit in our objective function during the training process.

The major challenges we plan to address with this paper are approaches that could execute imitation learning while ensuring cost threshold constraint, which is revealed only via the observations of expert trajectories. Our key contributions are summarized as follows:

- First, we formulate the cost-constrained imitation learning problem which represents the challenge of imitation learning in cost-constrained environments (where cost signals are known).

- We propose three methods to address the cost-constrained imitation learning problems. First, we design a Lagrangian-based method utilizing a three-way gradient update to solve the cost-constrained imitation learning problem. We then provide a meta-gradient approach that is able to tune the Lagrangian penalties of the first approach to significantly improve the performance. Finally, we use a cost-violation-based alternating gradient approach which has a different gradient update depending on the current solution feasibility.

- To further validate the effectiveness of our proposed method, we conducted extensive evaluations in Safety Gym environments (Ray, Achiam, and Amodei 2019) and MuJoCo environments (Todorov, Erez, and Tassa 2012). Our numerical evaluations show that the ensemble of our three proposed approaches can effectively imitate expert behavior while satisfying cost constraints.

## Background and Related Work

In this section, we describe the two problem models of relevance in this paper, namely Constrained MDPs and Imitation Learning. We also briefly review related work.

## Constrained Markov Decision Process

Reinforcement Learning problems are characterized by an underlying Markov Decision Process (MDP), which is defined by the tuple $(\mathcal{S}, \mathcal{A}, \mathcal{R}, \mathcal{P})$. Where $\mathcal{S}$ represents the set of states, $\mathcal{A}$ represents the set of actions. The reward function, $\mathcal{R} : \mathcal{S} \times \mathcal{A} \mapsto \mathbb{R}$, provides a quantitative measure of how well the system is performing based on the current state and action. The transition function, $\mathcal{P} : \mathcal{S} \times \mathcal{A} \times \mathcal{S} \mapsto [0, 1]$, defines the probability of transitioning from one state to another, given the current state and action taken. Specifically, the probability of transitioning from state $s$ to $s'$, given that action $a$ is taken, is represented by $\mathcal{P}(s'|s, a)$. A feasible set of policies, denoted as $\Pi$, contains all possible policies that can be implemented in the system. The objective of MDP is to find an optimal policy, $\pi \in \Pi$, by maximizing the reward-based objective function, which is defined as follows:

$$\max_{\pi \in \Pi} \mathbb{E}_\pi [\sum_{t=0}^{\infty} \gamma^t r(s_t, a_t)]. \tag{1}$$

In this work, we examine the scenario in which agents aim to optimize their rewards while adhering to policy-based cost constraints. This leads to an extension of the traditional MDP framework referred to as the Constrained Markov Decision Process (CMDP) (Altman 1999). The objective in a CMDP problem is succinctly formulated as:

$$\max_{\pi \in \Pi} \mathbb{E}_\pi [\sum_{t=0}^{\infty} \gamma^t r(s_t, a_t)], \quad s.t. \quad \mathbb{E}_\pi [\sum_{t=0}^{\infty} \gamma^t d(s_t, a_t)] \leq d_0. \tag{2}$$

Where $d(s, a)$ is the cost associated with taking action $a$ in state $s$ and is independent of the reward function, $r(s, a)$. $d_0$ is the expected cost threshold for any selected policy. There have been numerous approaches proposed for solving Constrained MDPs (Satija, Amortila, and Pineau 2020; Pathmanathan and Varakantham 2023), when the reward and transition models are not known *a priori*.

## Imitation Learning

Methods of Reinforcement Learning require clearly defined and observable reward signals, which provide the agent with feedback on their performance. However, in many real-world scenarios, defining these rewards can be very challenging. Imitation learning, on the other hand, offers a more realistic approach by allowing agents to learn behavior in an environment through observing expert demonstrations, without the need for accessing a defined reward signal.

An effective method for addressing imitation learning challenges is Behavior Cloning (BC) (Bain and Sammut 1995). This approach utilizes the states and actions of a demonstrator as training data, allowing the agent to replicate the expert's policy (Pomerleau 1991). One of the advantages of this method is that it does not require the agent to actively interact with the environment, instead, it operates as a form of supervised learning, similar to classification or regression. Despite its simplicity, BC is known to suffer from a significant drawback: the compounding error caused by covariate shift (Ross, Gordon, and Bagnell 2011). This occurs when

minor errors accumulate over time, ultimately resulting in a significantly different state distribution.

Another approach, Inverse Reinforcement Learning (IRL) (Ng, Russell et al. 2000) aims to identify the underlying reward function that explains the observed behavior of an expert. Once the reward function is determined, a standard Reinforcement Learning algorithm can be used to obtain the optimal policy. The reward function is typically defined as a linear (Ng, Russell et al. 2000; Abbeel and Ng 2004) or convex (Syed, Bowling, and Schapire 2008) combination of the state features, and the learned policy is assumed to have the maximum entropy (Ziebart et al. 2008) or maximum causal entropy (Ziebart, Bagnell, and Dey 2010). However, many IRL methods are computationally expensive and may produce multiple possible formulations for the true reward function. To address these challenges, Generative Adversarial Imitation Learning (GAIL) (Ho and Ermon 2016) was proposed. GAIL directly learns a policy by using a discriminator to distinguish between expert and learned actions, with the output of the discriminator serving as the reward signal. A more recent method, known as Inverse soft-Q Learning (IQ-Learn) (Garg et al. 2021), takes a different approach by learning a single Q-function that implicitly represents both the reward and policy, thereby avoiding the need for adversarial training. With their state-of-the-art performance in various applications, we designate GAIL and IQ-learn as baselines for our algorithms.

We now describe the imitation learning problem and GAIL approach here as it serves as the basis for our method. The learner's goal is to find a policy, denoted as $\pi$, that performs at least as well as an expert policy, denoted as $\pi_E$, with respect to an unknown reward function, denoted as $r(s, a)$. For a given policy $\pi \in \Pi$, we define its occupancy measure, denoted as $\rho_\pi \in \Gamma$, as (Puterman 2014)

$$\rho_\pi(s, a) = \pi(a|s) \sum_{t=0}^{\infty} \gamma^t P(s_t = s | \pi)$$

The occupancy measure represents the distribution of state-action pairs that an agent encounters when navigating the environment with the specified policy $\pi$. It is important to note that there is a one-to-one correspondence between the set of policies, $\Pi$, and the set of occupancy measures, $\Gamma$. Therefore, an imitation learning problem can be equivalently formulated as a matching learning problem between the occupancy measure of the learner's policy, $\rho_\pi$, and the occupancy measure of the expert's policy, $\rho_{\pi_E}$. In general, the objective can be succinctly represented as the task of finding a policy that closely matches the occupancy measure of the expert's policy, which is represented as:

$$\min_\pi -H(\pi) + \psi^*(\rho_\pi - \rho_{\pi_E}), \tag{3}$$

where $H(\pi) \triangleq \mathbb{E}_\pi[-\log \pi(a|s)]$ is the causal entropy of the policy $\pi$, which is defined as the expected value of the negative logarithm of the probability of choosing an action $a$ given a state $s$, under the distribution of the policy $\pi$. Additionally, the distance measure between the state-action distribution of the policy $\pi$, represented by $\rho_\pi$, and the expert's state-action distribution, represented by $\rho_{\pi_E}$, is represented by the symbol $\psi^*$. Specifically, the distance measure

(Jensen-Shannon divergence) employed by the GAIL framework is defined as follows:

$$\psi^*(\rho_\pi - \rho_{\pi_E}) = \max_D \mathbb{E}_\pi[\log D(s, a)] + \mathbb{E}_{\pi_E}[\log(1 - D(s, a))] \tag{4}$$

The GAIL method utilizes a combination of imitation learning and generative adversarial networks, where $D \in (0, 1)^{\mathcal{S} \times \mathcal{A}}$ acts as the discriminator. Through this formalism, the method trains a generator, represented by $\pi_\theta$, to generate state-action pairs that the discriminator attempts to distinguish from expert demonstrations. The optimal policy is achieved when the discriminator is unable to distinguish between the data generated by the generator and the expert data.

In our problem, as we aim to address the imitation learning problem within the constraints of an MDP, we have employed a unique distance measure that diverges from the traditional GAIL framework. This approach allows us to more effectively navigate the complexities of the constrained MDP setting and achieve our desired outcome.

## Lagrangian Method

In this section, we first describe the problem of cost-constrained imitation learning and outline our approach to compute the policy that mimics expert behaviors while satisfying the cost constraints.

We work in the $\gamma$-discounted infinite horizon setting, and we denote the expected reward and cost in association with policy $\pi \in \Pi$ as: $\mathbb{E}_\pi[r(s, a)] \triangleq \mathbb{E}_\pi[\sum_{t=0}^{\infty} \gamma^t r(s_t, a_t)]$ and $\mathbb{E}_\pi[d(s, a)] \triangleq \mathbb{E}_\pi[\sum_{t=0}^{\infty} \gamma^t d(s_t, a_t)]$, where $s_0 \sim p_0, a_t \sim \pi(\cdot|s_t)$, and $s_{t+1} \sim \mathcal{P}(\cdot|s_t, a_t)$ for $t \geq 0$. Formally, the Cost-Constrained Imitation Learning problem is a combination of the CMDP problem (2) and the Imitation Learning problem (3), and can be characterized as:

$$\min_\pi \quad -H(\pi) + \psi^*(\rho_\pi - \rho_{\pi_E})$$
$$s.t. \quad \mathbb{E}_\pi[d(s, a)] \leq \mathbb{E}_{\pi_E}[d(s, a)] \tag{5}$$

The imitation learning objective has two terms. The first term maximizes the entropy of the policy (to ensure no feasible alternative is excluded), while the second term minimizes the difference in occupation measures obtained using the policy and the expert trajectories. The constraint ensures that the expected cost computed using the policy is lower than the expected cost of the expert trajectories. In problems of interest, the cost function $d(s, a)$ is known but the reward is unknown.

Our generative approach to computing the policy that mimics the behavior of the expert within cost constraints relies on computing a solution to the unconstrained objective function of Equation (6) below. The theoretical justification for choosing this objective function is provided in the Appendix, and our theoretical analysis is based on the GAIL framework. Equation (6) is composed of three optimizations:

- Minimize the distance between state, action distributions of the new policy, $\pi_\theta$, and expert policy, $\pi_E$. This is transformed into the loss associated with a discriminator,

$D_\omega$, which discriminates between (state, action) pairs from experts, and (state, action) pairs generated by the new policy $\pi_\theta$.

- Maximize the entropy of the new policy, $\pi_\theta$, to ensure none of the correct policies are ignored.
- Minimize the cost constraint violations corresponding to the new policy, $\pi_\theta$.

$$
\begin{aligned}
L(\omega, \lambda, \theta) &\triangleq \min_\theta \max_{\omega, \lambda} \mathbb{E}_{\pi_\theta}[\log D_\omega(s,a)] + \\
&\mathbb{E}_{\pi_E}[\log(1 - D_\omega(s,a))] + \\
&\lambda \left( \mathbb{E}_{\pi_\theta}[d(s,a)] - \mathbb{E}_{\pi_E}[d(s,a)] \right) - \beta H(\pi_\theta),
\end{aligned} \quad (6)
$$

where $\theta$ represents the parameters of the policy, $\beta$ is the parameter corresponding to the causal entropy (since we maximize entropy similar to imitation learning) and finally, $\lambda$ is the Lagrangian multiplier corresponding to the cost constraint. $H(\pi_\theta) \triangleq \mathbb{E}_{\pi_\theta}[-\log \pi_\theta(a|s)]$ is the casual entropy of policy $\pi_\theta$. The given expert policy is represented by $\pi_E$, and a known cost function, represented by $d$, is also incorporated into the objective function.

Given the three optimization components, we do not choose one of the three but instead compute a saddle point $(\theta, \omega, \lambda)$ for (6). To accomplish this, we employ a parameterized policy, represented by $\pi_\theta$, with adjustable weights $\theta$, as well as a discriminator network, represented by $D_\omega$, which maps states and actions to a value between 0 and 1, and has its own set of adjustable weights $\omega$. The Lagrangian multiplier, denoted by $\lambda$, is for penalizing the number of cost constraint violations.

To obtain the saddle point, we update the parameters of policy, discriminator, and Lagrangian multiplier sequentially:

**Updating $\omega$:** The gradient of (6) with respect to $\omega$ is calculated as:

$$
\begin{aligned}
\nabla_\omega L(\omega, \lambda, \theta) &= \mathbb{E}_{\pi_\theta}[\nabla_\omega \log D_\omega(s,a)] + \\
&\mathbb{E}_{\pi_E}[\nabla_\omega \log(1 - D_\omega(s,a))]
\end{aligned} \quad (7)
$$

We utilize the Adam gradient step method (Kingma and Ba 2014) on the variable $\omega$, targeting the maximization of (6) in relation to $D$.

**Updating $\theta$:** To update policy parameters, we adopt the Trust Region Policy Optimization (TRPO) method (Schulman et al. 2015a). The theoretical foundation of the TRPO update process involves utilizing a specific algorithm to improve the overall performance of the policy by optimizing the parameters within a defined trust region:

$$
\theta_{k+1} = \arg\max_\theta \mathcal{L}(\theta_k, \theta), \quad s.t. \quad \bar{D}_{KL}(\theta||\theta_k) \le \delta. \quad (8)
$$

The key challenge in applying the TRPO update process is the computation of the surrogate advantage, denoted by $\mathcal{L}(\theta_k, \theta)$. It is a metric that quantifies the relative performance of a new policy $\pi_\theta$ in comparison to an existing policy $\pi_{\theta_k}$, based on data collected from the previous policy:

$$
\mathcal{L}(\theta_k, \theta) = \mathbb{E}_{s,a \sim \pi_{\theta_k}} \left[ \frac{\pi_\theta(a|s)}{\pi_{\theta_k}(a|s)} (A_r^{\pi_{\theta_k}}(s,a) - \lambda A_d^{\pi_{\theta_k}}(s,a)) \right] \quad (9)
$$

We do not have a reward function to compute the advantage and hence we utilize the output of the discriminator, represented by $-\log D_\omega(s,a)$, as the reward signal. Subsequently, we employ the Generalized Advantage Estimation (GAE) method outlined in Schulman et al. (2015b) to calculate the advantage of the reward, $A_r^{\pi_{\theta_k}}(s,a)$. Additionally, we also calculate the advantage pertaining to cost, denoted as $A_d^{\pi_{\theta_k}}(s,a)$, by utilizing the GAE method, as we know the cost function.

The average KL-divergence, represented by $\bar{D}_{KL}(\theta||\theta_k)$, between policies across states visited by the previous policy, can be computed as:

$$
\bar{D}_{KL}(\theta||\theta_k) = \mathbb{E}_{s \sim \pi_{\theta_k}} D_{KL}\left(\pi_\theta(\cdot|s)||\pi_{\theta_k}(\cdot|s)\right) \quad (10)
$$

**Updating $\phi_r, \phi_d$:** We do an Adam gradient step on $\phi_r$(reward value network parameters), $\phi_d$ (cost value network parameters)to minimize the mean-squared error of reward value and cost value, as we minimize these two loss functions:

$$
\begin{aligned}
\min_{\phi_r} \mathbb{E}_{s \sim \pi_{\theta_k}} (V_{\phi_r}^r(s_t) - \hat{R}_t^r)^2 \\
\min_{\phi_d} \mathbb{E}_{s \sim \pi_{\theta_k}} (V_{\phi_d}^d(s_t) - \hat{R}_t^d)^2
\end{aligned} \quad (11)
$$

Here $\hat{R}_t^r, \hat{R}_t^d$ are the reward to go and cost to go, which are calculated by the GAE method.

**Updating $\lambda$:** We do an Adam gradient step on $\lambda$ to increase (6), the gradient of (6) with respect to $\lambda$ is calculated as:

$$
\nabla_\lambda L(\omega, \lambda, \theta) = (\mathbb{E}_{\pi_\theta}[d(s,a)] - \mathbb{E}_{\pi_E}[d(s,a)]) \quad (12)
$$

Algorithm 1 shows the pseudocode for our approach, CCIL (Cost ConstraIned Lagrangian).

## Meta-Gradient for Lagrangian Approach

In this section, we introduce a meta-gradient approach to improve the Lagrangian method introduced in the previous section by applying cross-validation to optimize the Lagrangian multipliers. We call this approach MALM.

Meta-gradient is a strategy designed for the optimization of hyperparameters, such as the discount factor and learning rates in Reinforcement Learning problems. This approach involves the simultaneous execution of online cross-validation while pursuing the optimization objective of reinforcement learning, such as the maximization of expected return (Xu, van Hasselt, and Silver 2018; Calian et al. 2020). The goal is to optimize both inner and outer losses. The update of parameters on the inner loss is to update the parameters of the policy. The outer loss can be based on measures such as policy gradient loss and temporal difference loss, and is optimized by updating hyperparameters (Calian et al. 2020) in constrained RL problems, and using Distributed Distributional Deterministic Policy Gradients (D4PG) (Barth-Maron et al. 2018) algorithm framework. The critic loss is used in Barth-Maron et al. (2018) as the outer loss function to optimize the hyperparameters.

Instead of optimizing the hyperparameters, the key idea of MALM is to update the Lagrangian multiplier such that there is a better balance between reward maximization and

Algorithm 1: Cost ConstraIned Lagrangian (CCIL)
___

**Input**: initial parameters of policy $\theta$, reward value network $\phi_r$, cost value network $\phi_d$, discriminator network $\omega$, batch size $K$, a set of expert trajectories $\Phi_E = \{\tau_E \sim \pi_E\}$, initial Lagrangian multipliers $\lambda$, entropy parameter $\beta$, learning rates $\alpha_r, \alpha_d, \alpha_\lambda, \alpha_\omega$.

**Output**: Optimal policy $\pi_\theta$

1: Compute the average cost of expert trajectories: $J_E = \frac{1}{|\Phi_E|} \sum_{\tau \in \Phi_E} \sum_{t=1}^{T} d_t$

2: **for** $k = 1, 2, ...$ **do**

3:    Collect set of learner's trajectories $\Phi_k = \{\tau_i\}$ by running policy $\pi_{\theta_k}$ for $K$ time steps.

4:    Collect the reward $r_t$ of $K$ time steps by using the discriminator output: $r_t = -\log(D_\omega(s_t, a_t))$

5:    Compute $V_{\phi_r}^r(s_t)$ and $V_{\phi_d}^d(s_t)$ of $K$ time steps.

6:    Compute the reward and cost advantage $A^r(s_t, a_t)$ and $A^d(s_t, a_t)$, reward to go $\hat{R}_t^r$ and cost to go $\hat{R}_t^d$ of $K$ time steps by using GAE.

7:    Compute the average episode cost of learner's trajectories: $J_k = \frac{1}{|\Phi_k|} \sum_{\tau \in \Phi_k} \sum_{t=1}^{T} d_t$

8:    Update policy by using TRPO rule:
      $\theta' = \arg\max_\theta \sum_{t=1}^{K} \frac{\pi_\theta(a_t|s_t)}{\pi_{\theta_k}(a_t|s_t)}(A^r(s_t,a_t) - \lambda A^d(s_t,a_t)) + \beta H(\pi_{\theta_k})$

9:    Update reward value network:
      $\phi_r' \leftarrow \phi_r - \frac{1}{K}\sum_{t=1}^{K} \alpha_r \nabla_{\phi_r}(V_{\phi_r}^r(s_t) - \hat{R}_t^r)^2$

10:   Update cost value network:
      $\phi_d' \leftarrow \phi_d - \frac{1}{K}\sum_{t=1}^{K} \alpha_d \nabla_{\phi_d}(V_{\phi_d}^d(s_t) - \hat{R}_t^d)^2$

11:   Update discriminator network:
      $\omega' \leftarrow \omega + \frac{1}{K}\sum_{t=1}^{K} \alpha_\omega (\nabla_\omega[\log(D_\omega(s_t,a_t))] + \nabla_\omega[\log(1 - D_\omega(s_t,a_t))])$

12:   Update Lagrangian multipliers:
      $\lambda' \leftarrow \lambda + \alpha_\lambda(J_k - J_E)$

13:   $\theta \leftarrow \theta', \phi_r \leftarrow \phi_r', \phi_d \leftarrow \phi_d', \omega \leftarrow \omega', \lambda \leftarrow \lambda'$.

14: **end for**
___

cost constraint enforcement. Specifically, we use the outer loss that is defined as follows:

$$L_{outer}(\lambda) = \mathbb{E}_{\pi_\theta}(A^r(s,a) - \lambda d(s,a))^2 \quad (13)$$

Every batch is divided into training and validation data sets. The parameter update equations for the training data set are the same as described in the Lagrangian-based method (Equations (8) – (12)), and for the validation data set, we update the Lagrangian multiplier by minimizing the above outer loss function. MALM is similar to Algorithm 1 except that from lines 7 to 12, we update policy parameters, discriminator parameters, and Lagrangian multipliers based on the training data set. We also have an additional procedure to update the Lagrangian multipliers by minimizing the outer loss function based on the validation data set (see the Appendix for detailed pseudocode).

## Cost-Violation-based Alternating Gradient

We now describe our third method, Cost-Violation-based Alternating Gradient (CVAG), which does not rely on Lagrangian multipliers. Like previous methods, this method

also maintains a policy network, $\theta$, reward value network, $\phi_r$, and cost value network, $\phi_d$. The key novelty of this approach is in doing a feasibility check-based gradient update that is fairly intuitive. If the average episode cost of the learner does not exceed the average episode cost of experts (cost constraint), then we update the policy parameters towards the direction of maximizing the return, which would be the following equation in TRPO:

$$\mathcal{L}(\theta_k, \theta) = \max_\theta \mathbb{E}_{s,a \sim \pi_{\theta_k}}\left[\frac{\pi_\theta(a|s)}{\pi_{\theta_k}(a|s)}A_r^{\pi_{\theta_k}}(s,a)\right] \quad (14)$$

Otherwise, we update the policy towards the direction of minimizing the costs.

$$\mathcal{L}(\theta_k, \theta) = \min_\theta \mathbb{E}_{s,a \sim \pi_{\theta_k}}\left[\frac{\pi_\theta(a|s)}{\pi_{\theta_k}(a|s)}A_d^{\pi_{\theta_k}}(s,a)\right] \quad (15)$$

The detailed pseudocode is provided in the Appendix.

## Experiments

Table 1: Environments and expert trajectories.

| Task | Observation Space | Action Space | Dataset Size | Reward | Cost | Safety Coefficient |
|---|---|---|---|---|---|---|
| PointGoal1 | 60 | 2 | 10 | 18.77 ± 4.64 | 51.1± 3.36 | NA |
| PointButton1 | 76 | 2 | 10 | 18.56± 3.83 | 93.5 ± 7.8 | NA |
| CarGoal1 | 72 | 2 | 10 | 25.73 ±2.44 | 45.2 ± 6.35 | NA |
| CarButton1 | 88 | 2 | 10 | 11.84± 1.36 | 196.6 ± 25.44 | NA |
| DoggoGoal1 | 104 | 12 | 10 | 4.62± 1.55 | 57.9 ± 9.46 | NA |
| DoggoButton1 | 120 | 12 | 10 | 3.32± 1.2 | 181.7 ± 15.48 | NA |
| HalfCheetah-v3 | 17 | 6 | 10 | 4132.72±132.82 | 547.5 ± 8.1 | 0.4 |
| Hopper-v3 | 11 | 3 | 10 | 3594.04±1.78 | 433.6 ± 2.24 | 0.001 |
| Ant-v3 | 27 | 8 | 10 | 1263.43±142.14 | 653.2 ± 78.82 | 2 |
| Swimmer-v3 | 8 | 2 | 10 | 110.47±1.01 | 63.8 ± 1.08 | 1 |
| Walker2d-v3 | 17 | 6 | 10 | 1781.42± 19.06 | 130.1 ± 21.1 | 1 |
| Humanoid-v3 | 376 | 17 | 10 | 1744.53±257.22 | 206.6 ± 29 | 0.2 |

In this section, we compare our approaches against leading approaches for imitation learning (IL), including GAIL (Ho and Ermon 2016), IQ-Learn (Garg et al. 2021), Behavioral Cloning (BC) (Bain and Sammut 1995), and LGAIL (Cheng et al. 2023) in cost-constrained environments. This is to illustrate that a new approach is needed to mimic expert behaviors when there are unknown cost constraints. As we will soon demonstrate, all baselines suffer from either extensive cost constraint violations or low reward.

### Setup

**Environments and Cost Definition** We selected Safety Gym (Ray, Achiam, and Amodei 2019) and MuJoCo (Todorov, Erez, and Tassa 2012), two well-known environments from the literature for our evaluation purpose.

The Safety Gym environment is a standard platform for evaluating constrained RL algorithms. Each instance of this environment features a robot tasked with navigating a cluttered space to achieve a goal, all while adhering to specific constraints governing its interactions with objects and surrounding areas. In our experiments, there are three robotic agents (Point, Car, and Doggo) and two task types (Goal and Button), resulting in a total of six unique scenarios. The difficulty level for these environments is standardized to 1. Throughout each timestep, the environment provides distinct cost signals for various unsafe elements, each linked

Table 2: Overall performance of different environments. Normalized penalized return $R_{pen}$ captures the trade-off between achieving higher rewards and making the episode cost go below the expert's episode cost, higher is better. Recovered return $R_{rec}$ evaluates the extent that the agent imitates the expert's behavior, higher $R_{rec}$ means the agent imitated the experts better. Cost-Vio captures the extent that agent's episode cost goes beyond the expert's episode cost, lower is better.

| Task | | BC | GAIL | IQ-learn | LGAIL | CCIL | MALM | CVAG |
|---|---|---|---|---|---|---|---|---|
| CarGoal1 | $R_{pen}$ | 0.54±0.03 | -0.86± 0.34 | -0.57± 0.5 | -0.28±0.62 | 0.17±0.6 | **0.67± 0.08** | 0.61±0.06 |
| | $R_{rec}$ | 53.91 | 73.57 | 0 | 72.06 | **76.02** | 67.39 | 61.13 |
| | Cost-Vio | **0** | 20.55 | **0** | 6.47 | **0** | **0** | **0** |
| CarButton1 | $R_{pen}$ | 0.35±0.45 | -0.59± 0.07 | 0.05± 0.02 | -0.36±0.47 | 0.13±0.53 | 0.01±0.52 | **0.42±0.08** |
| | $R_{rec}$ | 56.84 | **60.73** | 5.49 | 59.29 | 58.19 | 51.94 | 42.48 |
| | Cost-Vio | **0** | 18.01 | **0** | 10.09 | **0** | **0** | **0** |
| PointGoal1 | $R_{pen}$ | 0.81±0.03 | 0.35± 0.78 | -1.07± 0.91 | 0.1±0.72 | 0.07± 0.73 | **0.95±0.09** | 0.87±0.11 |
| | $R_{rec}$ | 80.87 | 95.05 | 2.82 | **95.15** | 93.55 | 94.78 | 87.32 |
| | Cost-Vio | **0** | 5.39 | 12.3 | 7.35 | 3.19 | **0** | **0** |
| PointButton1 | $R_{pen}$ | -0.19±0.53 | -0.55± 0.06 | -1.26± 0.21 | 0.37±0.54 | -0.19±0.54 | **0.44±0.53** | 0.27±0.48 |
| | $R_{rec}$ | 69.5 | **73.49** | 7.44 | 68.53 | 69.88 | 72.95 | 55.98 |
| | Cost-Vio | 5.04 | 15.41 | 19.71 | **0** | 4.42 | **0** | **0** |
| DoggoGoal1 | $R_{pen}$ | -0.3 ±0.57 | -0.94±0.03 | 0± 0.01 | 0.25±0.04 | **0.36±0.07** | -0.25± 0.6 | 0.29 ± 0.09 |
| | $R_{rec}$ | 27.92 | **44.81** | 0 | 25.32 | 35.5 | 28.79 | 33.33 |
| | Cost-Vio | 0.47 | 14.97 | **0** | **0** | **0** | **0** | **0** |
| DoggoButton1 | $R_{pen}$ | 0.35±0.06 | -0.38± 0.55 | -0.01±0.04 | 0.14±0.11 | **0.35±0.05** | 0.3± 0.05 | 0.06± 0.1 |
| | $R_{rec}$ | 34.94 | **41.27** | -0.6 | 14.16 | 34.64 | 30.42 | 5.72 |
| | Cost-Vio | **0** | 1.96 | **0** | **0** | **0** | **0** | **0** |
| HalfCheetah-v3 | $R_{pen}$ | -6.16±0.08 | -5.91±0.05 | -7.08±0.1 | -0.29 ± 1.63 | 0.44±0.37 | **0.81±0.07** | 0.53 ± 0.26 |
| | $R_{rec}$ | 70.61 | **95.19** | 0 | 51.91 | 44.47 | 81.49 | 52.91 |
| | Cost-Vio | 392.94 | 392.12 | 417.49 | **0** | **0** | **0** | **0** |
| Hopper-v3 | $R_{pen}$ | 0.16±0.02 | 0.48±0.61 | 0.07± 0.03 | 0.02 ±0.49 | **1±0.01** | 0.75±0.48 | 0.5±0.59 |
| | $R_{rec}$ | 16.11 | 98.78 | 6.54 | **99.92** | 99.71 | 99.62 | 98.58 |
| | Cost-Vio | **0** | 0.49 | **0** | 4.16 | **0** | **0** | **0** |
| Ant-v3 | $R_{pen}$ | -0.73±0.18 | -0.69±0.02 | 0.15± 0.07 | 0.77± 0.1 | 0.06±0.61 | **0.86±0.03** | 0.53± 0.47 |
| | $R_{rec}$ | 86.64 | **95.59** | 14.8 | 76.82 | 79.86 | 85.76 | 78.1 |
| | Cost-Vio | 213.94 | 242.25 | **0** | **0** | 3.12 | **0** | **0** |
| Swimmer-v3 | $R_{pen}$ | -2.37±2.02 | -0.17 ±0.56 | 0.36 ± 0.05 | -0.42± 0.12 | **0.7 ± 0.49** | 0.46 ± 0.6 | 0.63± 0.49 |
| | $R_{rec}$ | 58.44 | **94.87** | 35.71 | 93.35 | 94.76 | 94.29 | 87.41 |
| | Cost-Vio | 93.13 | 8.57 | **0** | 7.96 | **0** | **0** | **0** |
| Walker2d-v3 | $R_{pen}$ | -0.34±0.06 | -0.06±0.53 | 0.19±0.06 | 0.2±0.63 | 0.73±0.5 | 0.72±0.51 | **0.97±0.01** |
| | $R_{rec}$ | 99.2 | **99.35** | 19.04 | 98.57 | 98.53 | 98.58 | 97.25 |
| | Cost-Vio | 14.02 | 8.1 | **0** | **0** | **0** | **0** | **0** |
| Humanoid-v3 | $R_{pen}$ | 0.32±0.01 | -0.26±0.02 | 0.21±0.02 | 0.86 ± 0.02 | **0.95 ± 0.02** | 0.39±0.52 | 0.82± 0.03 |
| | $R_{rec}$ | 32.21 | **122.94** | 20.6 | 86.01 | 94.83 | 95.77 | 82.4 |
| | Cost-Vio | **0** | 73.54 | **0** | **0** | **0** | **0** | **0** |

to an associated constraint. Additionally, an aggregate cost signal is provided, encapsulating the overall impact of the agent's interactions with unsafe elements. The cost functions are straightforward indicators, evaluating whether an unsafe interaction has occurred ($c_t = 1$ if the agent engages in an unsafe action, otherwise $c_t = 0$).

In order to illustrate the robustness of our methods, we also adopt MuJoCo for its wide array of continuous control tasks, such as Walker2d and Swimmer. These tasks are frequently employed to assess RL and IL algorithms. For reward, we utilized the default MuJoCo environment settings, however, as the MuJoCo environment does not have built-in cost constraints like in Safety Gym, we introduce constraints on features of the state space and action space:

- Control Cost. The agent faces penalties for taking excessively large actions in HalfCheetah-v3 and Hopper-v3.
- Control Cost plus Contact Cost. This is utilized in Ant-v3 and Humanoid-v3. The contact cost of an agent is generated if the external contact force is too large, the cost indicator is the sum of control cost and contact cost.
- Speed Limit: This is utilized in Swimmer-v3 and Walker2d-v3. An agent is penalized when it moves at a much higher speed.

*Safety coefficient* is the cost threshold for the cost indicator. At each time step, if the cost indicator is larger than

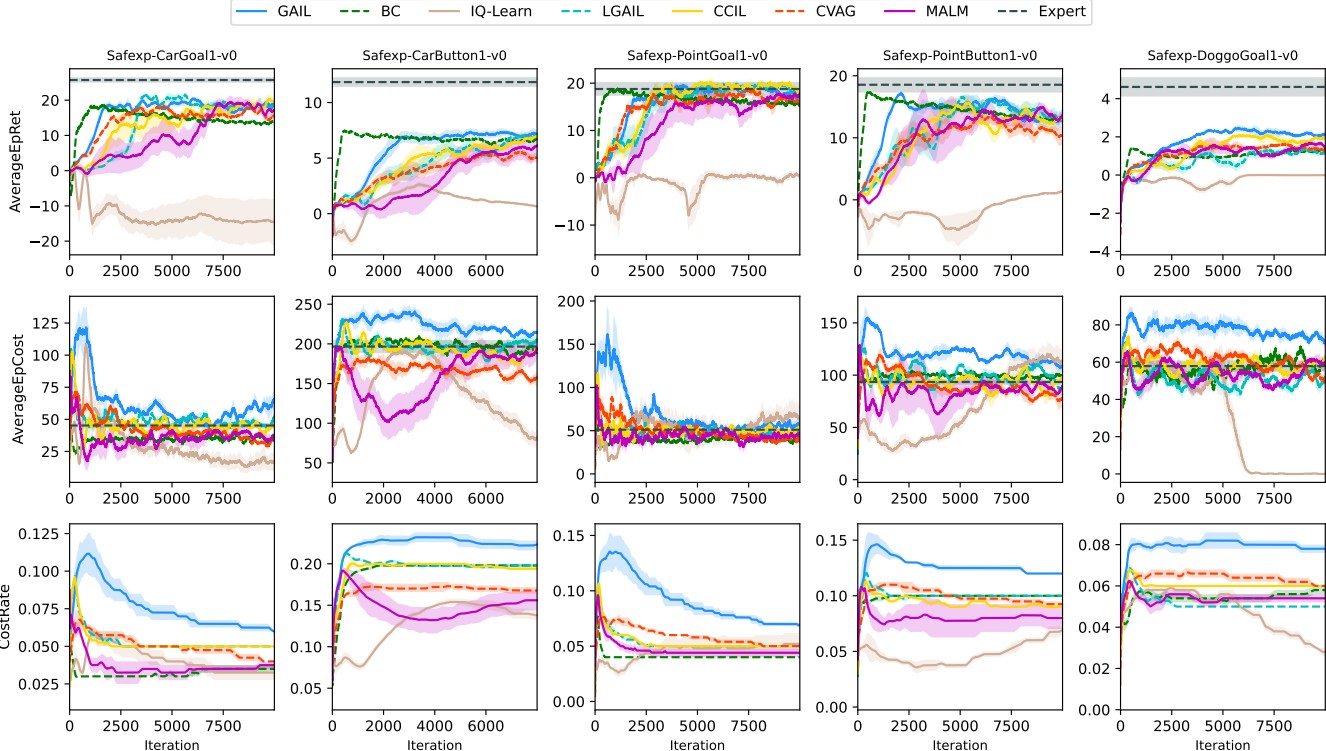

Figure 1: Performance of Safety Gym environments. The x-axes indicate the number of iterations, and the y-axes indicate the performance of the agent, including average rewards/costs/cost rates with standard deviations.

the *safety coefficient*, then the agent will get a cost of 1, otherwise, the cost is denoted as 0. The expert trajectories were generated by solving a forward-constrained RL problem, and the statistics of these trajectories and Safety Coefficient parameters are summarized in Table 1.

**Baselines and Code**. In order to evaluate the performance of our algorithms, we performed a comparison with three popular IL methods without explicit cost constraints consideration (GAIL, IQ-Learn, and BC). For the constraint-aware IL method, we use LGAIL as the baseline. As noted earlier, LGAIL assumes the knowledge of cost limit $d'$, which is used as a parameter in their objective function:

$$\min_\theta \max_{\omega,\lambda} \mathbb{E}_{\pi_\theta}\left[\log D_\omega(s,a)\right] + \mathbb{E}_{\pi_E}\left[\log(1 - D_\omega(s,a))\right] +$$

$$\lambda \left(\mathbb{E}_{\pi_\theta}[d(s,a)] - d'\right) - \beta H(\pi_\theta). \quad (16)$$

As they defined $d'$ to be less than the minimal episode cost of expert trajectories, so we use the $90\%$ of minimal episode cost of expert trajectories as $d'$. In the case of BC, the expert trajectories dataset, which consists of state-action pairs, was divided into a $70\%$ training data set and a $30\%$ validation data set. The policy was then trained using supervised learning techniques. Then in the GAIL method, the policy network, reward value network, cost value network, and discriminator network all employ the same architectures, comprising two hidden layers of 100 units each, with tanh nonlinearities being utilized in the layers. Lastly, for the IQ-Learn method, we use the same setting as illustrated in (Garg

et al. 2021), that we use critic and actor networks with an MLP architecture with 2 hidden layers and 256 hidden units.

**Implementation**. We employ a neural network architecture consistent with the one utilized in the GAIL method. In addition, our approaches add a cost value network and a Lagrangian penalty term $\lambda$ (in CCIL and MALM), which distinguishes our method from GAIL. The policy, value, and cost value network are optimized through gradient descent with the Adam optimizer (Kingma and Ba 2014). The initial value of $\lambda$ is set to 0.01 and optimized using the Adam optimizer. In MALM, the state-action pairs of each batch size are divided into a $70\%$ training data set and a $30\%$ validation data set. We run each algorithm for 5 different random seeds. The algorithms ran for 2000 time steps (batch size) during each iteration, and the episode's total true reward and cost are recorded. The details of the hyper-parameters used in the experiments can be found in the Appendix. The implementation of all code is based on the OpenAI Baselines library (Dhariwal et al. 2017).

**Performance Metrics**. We use different performance metrics to compare overall performance. Firstly, followed by (Ray, Achiam, and Amodei 2019), we record the average episode's true reward, the average episode's cost, and the cost rate over the entirety of training (the sum of all costs divided by the total number of environment interaction steps) throughout the training. We also incorporate the normalized *penalized return* introduced by (Calian et al. 2020). This metric effectively captures the delicate balance between

maximizing rewards and ensuring that the episode cost remains below the expert's episode cost. The formulation is represented as $R_{pen} = R/R_E - \mathcal{K}\max(J_d^{\pi}/J_d^{\pi_{\mathbb{E}}} - 1, 0)$, where $R$ and $J_d^{\pi}$ denote the average episode true reward/cost for the algorithm upon convergence (computed as an average over the last 100 iterations), and $R_E$, $J_d^{\pi_{\mathbb{E}}}$ represent the average episode reward/cost of the expert. The second term in the equation introduces a penalty if the episode cost exceeds the expert cost; otherwise, this term remains zero. The constant $\mathcal{K}$ serves as a fixed parameter determining the weight assigned to the constraint violation penalty. To effectively penalize algorithms consistently breaching cost constraints during evaluation, we set $\mathcal{K} = 1.2$. The *recovered return* metric captures the degree to which the agent replicates the expert's behavior, denoted as $R_{rec} = R/R_E \times 100$. A value of $R_{rec} \geq 100$ signifies proficient imitation of the expert, while an approach to 0 indicates the agent's inability to replicate the expert's behavior (with $R_{rec}$ being 0 when returns are negative). Finally, *cost violation* is defined as $\phi_d = \max(0, J_d^{\pi} - J_d^{\pi_{\mathbb{E}}})$. In cases where the agent's episode cost is less than the expert's episode cost, denoting no cost violation, $\phi_d$ is set to 0.

## Results

In order to assess the effectiveness of each algorithm, we used average episode true reward, average episode cost, and cost rate at each iteration as an evaluation. Here are key observations from Figures 1 (Training curves for these three evaluation metrics in Mujoco and DoggoButton tasks can be found in the Appendix):

- In the BC method, which doesn't consider cost constraints, three key observations stand out. First, in tasks like Hopper, both reward and cost remain consistently low. Second, while the reward tends to approach expert levels, the cost often surpasses the constraint or the expert's cost during training. Lastly, there's a pattern where the average return initially rises but later drops in tasks like CarGoal and PointButton. This might be due to BC being prone to overfitting. Initially, the model fits the training data well, boosting performance. However, as training continues, the model can become too specialized, struggling to handle new situations. Additionally, the cost rate in this method is relatively high compared to other approaches.
- The GAIL method, which also overlooks cost constraints, exhibited a pattern wherein the reward approached the expert reward during training. However, the cost consistently exceeded the expert's cost throughout the training stage, and the cost rate was notably the highest across most environments.
- For the IQ-Learn method, the cost of the policy obtained was generally lower than that of the expert cost in most tasks. However, the reward obtained was well below the expert reward.
- The LGAIL method can achieve performance close to the expert's reward in certain tasks, demonstrating competitive efficacy compared to CCIL. However, it tends to incur higher costs and cost rates throughout the training

process.
- Our approach, MALM, demonstrated superior performance compared to all other methods, consistently offering the optimal balance of reward and adherence to cost constraints. Notably, in the HalfCheetah task, MALM excelled in approximating expert behavior, outperforming other methods that exhibited a trend of decreasing rewards or surpassing the expert's cost. Following MALM, CVAG, and CCIL secured the second-best performance, surpassing GAIL, BC, and IQ-Learn in terms of both reward and cost.

Table 2 presents a performance comparison of various methods based on *penalized return*, *recovered return*, and *cost violation*. In terms of these metrics, we observed that while the GAIL method can achieve the highest recovery of experts' return in most environments, it often violates cost constraints. Our methods consistently attained the highest penalized return and exhibited no cost violation in the majority of tasks. LGAIL also violated cost constraints in certain tasks. Although BC can recover a high percentage of experts' returns in some tasks, it tends to violate cost constraints. IQ-Learn falls short in competitive performance in recovering expert's returns and avoiding exceeding experts' episode costs. In comparison, our methods demonstrate superior optimization in both cost and reward aspects.

## Conclusion

In this study, we address a novel challenge of solving the imitation learning problem within cost constraints. To tackle this issue, we provide three methods that are both scalable and effective. First, we provide a Lagrangian method to solve the cost-constrained imitation learning problem. We then provide a meta-gradient approach that is able to tune the Lagrangian penalties of the first approach to significantly improve the performance. Finally, we provide a cost violation-based alternating gradient approach that has a different gradient update depending on the current solution's feasibility. Experiments demonstrate that our methods can effectively imitate expert behavior while satisfying cost constraints, compared to other imitation learning methods that do not consider cost constraints. Among our approaches, the meta-gradient approach achieved the best trade-off between achieving high expected rewards while satisfying the cost constraints.

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
