# OpenReview forum: "Imitating Cost Constrained Behaviors in Reinforcement Learning"
_icaps-conference.org/ICAPS/2024/Conference — ICAPS 2024_

### Official Review · Reviewer_8db6 · 2024-01-22

**Significance And Importance:** 2
**Soundness:** 3
**Novelty:** 2
**Clarity:** 3
**Overall Evaluation:** 1
**Confidence:** 3

**Weaknesses:**

1: Minor weaknesses that are easily fixable.

**Contributions Of The Paper:**

The main contribution of the paper is the development of three methods for performing imitation learning with cost constrains. This is relevant in scenarios in which agent decisions are influenced by both a reward model and cost constraints. The models are a Lagrangian-based method, a meta-gradient method, and cost-violoation-based Alternating Gradient. The paper also presents an empirical evaluation of these methods in diverse Safety Gym and MuJoCo scenarios.

**Ethical Considerations:**

(1) Not Applicable: The paper does not have any ethical considerations to address

**Nomination For Best Paper:**

No

**Questions For Authors:**

1) Lack of focus: The authors present three approaches, some of which build on each other. However, there is no detailed discussion of the approaches (e.g., which of the approaches is suitable for which scenarios?) or an experimental comparison with derived results (e.g., in which experiments did which approach work well and why?). As a result, the contribution is not clear and the transferability of the results is not clear to the reader. I recommend that the authors focus on one of the approaches (e.g. MALM) and discuss it in the paper in more detail.
2) State of the art and related work in cost-constrained reinforcement learning. The authors mention some approaches in imitation learning, especially those on which their approaches are based (e.g. LGAIL). However, I would be interested to know what is the state of research on cost-constrained RL? Why weren't some of the studies mentioned in the paper or used as a baseline? How do the authors distinguish their work from these approaches?
3) The authors state in section 1 that the approach of Cheng et al. (2023) is very similar to their approach, but that the authors "do not assume knowledge of the security constraints". This aspect seems to be an important difference between the approaches. However, the authors did not explicitly prove this later in the paper, did they? Could the authors explain how they addressed this in their experiments and whether they took this into account?
4) In the results section, the authors state that the MALM approach "demonstrated superior performance compared to all other methods, consistently offering the optimal balance of reward and adherence to cost constraints." I have issues with the statement. 1) What does "optimal balance" mean, could the authors clarify what exactly they mean by this statement? 2) I don't see the overperformance of MALM in the plots shown. There are many environments where MALM does not perform best according to the plots, e.g. CarGoal, PointGoal, DoggoGoal.  I recommend a more detailed discussion of the results here.
5) The authors barely address the results in Table 2, in particular I wonder how the results relate to the results in Figure 1. Overall, I miss a discussion of the results as a whole, especially since many approaches are compared. Three of them come from the authors, with a different approach always performing best. This leaves me wondering what to take away from the results.

Overall, the paper presents relevant and good results. If the authors address the weaknesses and comments, I belive that the quality of the paper can be improved.

**Reproducibility:**

3: Authors describe the implementation and domains in sufficient detail.

**Strengths Of The Paper:**

- Relevance: The paper addresses cost-constrained reinforcement learning, which is highly relevant in reinforcement learning research, especially in critical scenarios such as autonomous driving, where safety and cost are important aspects to consider during training.
- Benchmarking: the authors compare their approaches with state-of-the-art approaches in imitation learning in various gym scenarios.
- Detailed description of the approaches

**Weaknesses Of The Paper:**

- Lack of methodological focus (see comment 1)
- Lack of related work on cost-constrained reinforcement learning (see comment 2)
- Missing clarity of the experimental results (see comments 4 and 5)

---

> ### Author Rebuttal · Authors · 2024-01-28
>
> We thank the reviewer for all the comments. We will integrate our replies into our final paper if it's accepted.
> 1. It turns out that different approaches work well in different types of cost constraints. We provide a summary of this discussion later in Q5 below.
>
> We will modify and update the result discussion section to reflect these observations.
>
> 2. To the best of our knowledge, LGAIL is the SOTA framework for safe IL tasks. Cheng et al. (2023) have explored the literature thoroughly, and they view the safe IL tasks as a Constrained Markov Decision Process (CMDP). We will expand our related work section to reflect this part of the literature better.
>
> In terms of benchmarking baselines, we have chosen 4 most representative baselines: (1) BC: the baseline of all IL-related work; (2) GAIL/IQ Learn: SOTA for unconstrained IL tasks; and (3) LGAIL: current SOTA for cost-constrained IL tasks. We compare only against LGAIL for the cost-constrained IL methods, as LGAIL has already established itself as the SOTA through extensive experiments.
>
> 3. In the LGAIL method, Cheng et al. predefined the cost limit d_0, which can be interpreted as the threshold constraint on cost. For our approaches, we do not require such threshold constraints and instead rely solely on expert trajectories. In Appendix A, we have derived and proved the form of the final objective function which does not need the predefined d_0.
>
> 4. We will remove the original statement and replace it with a more thorough result analysis (see below).
>
> 5. The results presented in Table 2 are best explained by looking at the type of constraints. When agents are penalized for:
> - entering dangerous areas (PointGoal, CarGoal, DoggoGoal), MALM emerges as the best method overall, except DoggoGoal, in which CCIL ranks first.
> - entering dangerous areas and pressing the wrong buttons (PointButton, CarButton, DoggoButton), there is no clear winner (all 3 methods are ranked first in separate instances). However, from Figure 1, MALM curves demonstrate better reward and cost performance overall.
> - large actions, MALM is the best for HalfCheetah and CCIL is the best for Hopper.
> - large actions and large contact cost, MALM is the best for Ant and CCIL is the best for Humanoid.
> - exceeding speed limits (Swimmer and Walker2d), CVAG is the better choice (best in Walker2d, 2nd in Swimmer).

---

### Official Review · Reviewer_rWyB · 2024-01-22

**Significance And Importance:** 2
**Soundness:** 3
**Novelty:** 3
**Clarity:** 4
**Overall Evaluation:** 2
**Confidence:** 3

**Weaknesses:**

1: Minor weaknesses that are easily fixable.

**Contributions Of The Paper:**

- Introduction of three novel methods for imitating expert behavior while satisfying cost constraints. Noting that cost constraints satisfaction has not been considered in previous related work.
- It also contributes with an interesting and comprehensive experimental validation in Safety Gym and MuJoCo environments.

**Ethical Considerations:**

(1) Not Applicable: The paper does not have any ethical considerations to address

**Nomination For Best Paper:**

No

**Questions For Authors:**

1. How can you check for scalability of your proposal?

2. What kind of real world application the authors can suggest for the proposed method?

**Reproducibility:**

2: Some details are missing, but the paper still appears to be replicable with some effort.

**Strengths Of The Paper:**

- Unique focus on cost-constrained imitation learning.
- Extensive experimental validation.
- Strong theoretical foundation detailed in supplemental material.

**Weaknesses Of The Paper:**

Absence of publicly available code impacting reproducibility.

---

> ### Author Rebuttal · Authors · 2024-01-28
>
> We thank the reviewer for the comments. Below please find our responses to the raised concerns:
>
> 1. Scalability?
> ---
> **[Ans]**
> The scalability of our method is best illustrated in Table 1, in which we can see a wide range of problem complexities. In terms of observation space, the dimension ranges from 8 to 376, while the action space dimensions range from 2 to 17. Even in the environment with the highest dimensionality, we did not observe any significant computational time increase.
>
> 2. Real world application?
> ---
> **[Ans]**
> Our paper falls in the "imitation learning" (IL) area, but with cost constraints. Similar to the general IL, our approach is best suited for domains where defining a reward function is challenging or a human expert's expertise is valuable yet hard to quantify. The most notable applications are in robotics, autonomous vehicles, and game playing, where human experts can easily balance between the goal to be optimized and the cost constraints to be observed.
>
> All numerical experiments in our paper are in the domain of robotics (Safety Gym and MuJoCo). These environments are chosen for benchmarking convenience (both are widely used in the literature); but our approach can be applied to other domains sharing similar set up.

---

### Official Review · Reviewer_wYZy · 2024-01-23

**Significance And Importance:** 2
**Soundness:** 3
**Novelty:** 2
**Clarity:** 3
**Overall Evaluation:** 1
**Confidence:** 4

**Weaknesses:**

0: Minor weaknesses requiring some work to be addressed for the paper to be accepted.

**Contributions Of The Paper:**

The paper considers the problem of Imitation Learning under cost constraints, in which it is assumed that the agent tries to maximize a reward function, while ensuring that constraints on costs are not violated. The article proposes three different methods to solve this problem: CCIL, MALM, and CVAG.

Experiments are presented in the SAFE GYM and Mujoco environment and compared with other alternatives in the literature. The experiments demonstrate that frameworks that do not take into account constraints on costs end up generating policies that violate such constraints, as it was expected.

**Ethical Considerations:**

(1) Not Applicable: The paper does not have any ethical considerations to address

**Nomination For Best Paper:**

No

**Questions For Authors:**

The problem addressed by LGAIL is very similar to the problem defined in the article. In such a way that LGAIL can be easily adapted to the problem in question. What is the difference between LGAIL algorithm and CCIL algorithm?

**Reproducibility:**

2: Some details are missing, but the paper still appears to be replicable with some effort.

**Strengths Of The Paper:**

Three different approaches to the problem are proposed, which improves on methods from literature.

The paper presents a good set of experiments.

**Weaknesses Of The Paper:**

Although three different approaches are presented, they are not a break through and follows direct from well-known paper from literature.

Although a good set of experiments are presented, there is no discussion or hint on which approach to use and when.

---

> ### Author Rebuttal · Authors · 2024-01-28
>
> We thank the reviewer for the comments. Below please find our responses to the raised concerns:
>
> 1. Although three different approaches are presented, they are not a break through and follows direct from well-known paper from literature. Although a good set of experiments are presented, there is no discussion or hint on which approach to use and when.
> ---
> **[Ans]**
> Reviewer's point is well taken and the results presented in Table 2 are best explained by looking at the type of constraints. When agents are penalized for:
> - entering dangerous areas (PointGoal, CarGoal, DoggoGoal), MALM emerges as the best method overall on average.
> - entering dangerous areas and executing wrong actions (PointButton, CarButton, DoggoButton), there is no clear winner (all 3 methods are ranked first in separate instances). However, from Figure 1, MALM curves demonstrate better performances in reward and cost overall on average.
> - big deviations in motion, MALM is the best for HalfCheetah and CCIL is the best for Hopper.
> - big deviations in motion and actions, MALM is the best for Ant and CCIL is the best for Humanoid.
> - exceeding speed limits (Swimmer and Walker2d), CVAG is the better choice (best in Walker2d, 2nd in Swimmer).
>
>
> 2. The problem addressed by LGAIL is very similar to the problem defined in the article. In such a way that LGAIL can be easily adapted to the problem in question. What is the difference between LGAIL algorithm and CCIL algorithm?
> ---
> **[Ans]** In the LGAIL method, Cheng et al. predefined the cost limit d_0, which can be interpreted as the "security constraints". For our approaches, we do not require such user inputs and instead rely solely on expert trajectories. In Appendix A, we have also derived and proved the form of the final objective function which does not need the predefined d_0.

---

### Meta-Review · Area_Chair_cufa · 2024-02-02

**Recommendation:** Accept (Poster)
**Confidence:** 3

**Metareview:**

The paper presents three new methods for imitation learning with cost constraints, namely a Lagrangian-based method, a meta-gradient method, and cost-violoation-based Alternating Gradient. The experimental evaluation uses Safety Gym and MuJoCo scenarios.

Reviewers agreed that the presented approaches are novel and the approached problem is interesting.
We recommend to include the rebuttal information in the camera-ready paper.

**Ethical Considerations:**

(1) Not Applicable: The paper does not have any ethical considerations to address